# Penalizing Length: Uncovering Systematic Bias in Quality Estimation Metrics

## Abstract

Quality Estimation (QE) metrics are vital in machine translation for reference-free evaluation and as a reward signal in tasks like reinforcement learning. However, the prevalence and impact of length bias in QE have been underexplored. Through a systematic study of top-performing regression-based and LLM-as-a-Judge QE metrics across 10 diverse language pairs, we reveal two critical length biases: First, QE metrics consistently over-predict errors with increasing translation length, even for high-quality, error-free texts. Second, they exhibit a preference for shorter translations when multiple candidates are available for the same source text. These inherent length biases risk unfairly penalizing longer, correct translations and can lead to sub-optimal decision-making in applications such as QE reranking and QE guided reinforcement learning. To mitigate this, we propose two strategies: (a) applying length normalization during model training, and (b) incorporating reference texts during evaluation. Both approaches were found to effectively reduce the identified length bias.

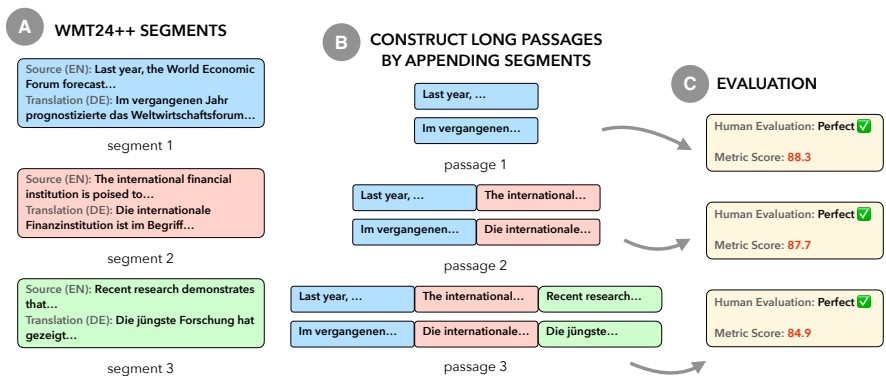

Figure 1: QE metrics exhibit a length-related penalty when evaluating long-form translations, as metric scores decrease when translations become longer despite being error-free. As shown, concatenating consecutive gold-standard segments from the same document leads to progressively lower predicted scores, in contrast to human evaluations.

## 1 Introduction

Quality Estimation (QE) metric is a statistical score that predicts the quality of machine-translated text without relying on a reference translation, making it particularly useful in scenarios where human-produced references are unavailable. Broadly, existing QE metrics fall into two categories: regression-based models, such as MetricX (Juraska et al., 2024) and COMET (Rei et al., 2020), which learn to directly assess the hypothesis from annotated training data, and LLM-as-a-Judge approaches, such as InstructScore (Xu et al., 2023) and AutoMQM (Fernandes et al., 2023), which leverage large language models to produce an error rating that approximates human judgments.

Length bias refers to the systematic tendency of evaluation models to prefer or penalize outputs depending on their length rather than their true quality. Prior research has documented such effects

in related evaluation settings, including LLM-as-a-Judge frameworks (Domhan & Zhu, 2025) and reward models used in reinforcement learning (Zhao et al., 2025). However, the extent to which QE metrics suffer from length bias remains largely underexplored. This gap is consequential, as length bias can distort system comparisons, over- or under-value longer translations, and ultimately misguide both model development and data curation.

In this study, we design two complementary experiments to systematically investigate the presence of length bias in QE metrics. The first experiment evaluates length invariance by concatenating high-quality, error-free segments into progressively longer translations, allowing us to assess whether metrics remain stable as input length increases. The second experiment directly compares alternative hypotheses generated from the same source, enabling us to measure whether metrics exhibit a consistent preference toward shorter or longer outputs. Our analyses reveal two concerning patterns: QE metrics tend to assign spurious errors as translations grow longer, even when no actual errors are present, and they disproportionately favor shorter hypotheses when multiple candidates exist. These findings highlight a fundamental vulnerability—QE metrics risk unfairly penalizing correct but longer translations, which can in turn misguide downstream applications such as system reranking and reinforcement learning pipelines that depend on reliable QE signals.

To address the shortcomings, we propose and evaluate two mitigation strategies aimed at reducing length bias in QE metrics. The first approach introduces length normalization during model training, encouraging regression-based QE models to disentangle translation quality from sequence length and thereby improving their robustness. The second approach supplements QE evaluation with reference-based metrics to provide an external quality anchor that counteracts length bias toward shorter outputs. Our experiments show that both strategies are effective in reducing length sensitivity and yield more reliable and equitable evaluations across translations of varying lengths.

## 2 PRELIMINARIES

### 2.1 QUALITY ESTIMATION FOR MACHINE TRANSLATION

**Task Definition.** Given a source text $x$ and a hypothesis translation $h$, Quality Estimation (QE) is the task of automatically predicting a quality score $Q(x, h)$ that reflects the quality of the hypothesis. A lower score indicates a lower translation quality.

**Evaluation.** Unlike reference-based evaluation, which assesses a hypothesis translation by measuring its similarity to one or more human references (e.g., BLEU (Papineni et al., 2002) and COMET (Rei et al., 2020)), QE operates without references and instead assesses the quality of a hypothesis directly against the source text. In general, QE models take $(x, h)$ as input and are trained to predict potential errors in $h$, then compute an overall quality score $Q(x, h)$ by aggregating these error ratings according to their type and severity.

Most modern QE metrics adopt the error annotation scheme defined in the Multidimensional Quality Metrics (MQM) framework (Freitag et al., 2021). MQM classifies each error into a type (e.g., accuracy, fluency, style, terminology, non-translation, or other) and assigns a severity level (e.g., minor, major, or critical). Let $Err(x, h) = \{e_1, \ldots, e_n\}$ denote the set of MQM-annotated errors in hypothesis $h$ given source $x$. Each error $e_i$ is characterized by its type $t(e_i)$ and severity $s(e_i)$, and its contribution is determined by a rating function $r(t(e_i), s(e_i))$. The overall error rating of $h$ is defined by

$$R(x, h) = \sum_{e_i \in Err(x,h)} r\big(t(e_i), s(e_i)\big). \tag{1}$$

Typically, minor and major errors are penalized with scores of $-1$ and $-5$, respectively, while more severe cases, such as non-translation, may incur penalties as large as $-25$.

Given $R(x, h)$, a QE metric defines a corresponding quality score $Q(x, h)$ either by direct negation (e.g., $Q(x, h) = -R(x, h)$) or by mapping it to a direct assessment (DA) scale (e.g., $Q(x, h) = 100 - 4R(x, h)$). Some regression-based QE metrics further constrain $Q(x, h)$ within a predefined range (e.g., $Q(x, h) \in [0, 25]$ for MetricX-24 QE (Juraska et al., 2024) and $Q(x, h) \in [0, 1]$ for CometKiwi (Rei et al., 2022)), which are often rescaled to ensure comparability across metrics.

## 2.2 APPLICATIONS OF QUALITY ESTIMATION

**Data Filtering.** Data filtering refers to the task of selecting high-quality source–translation pairs from large, noisy corpora (e.g., web-crawled data) for building or refining a training corpus. Given a set of candidates $(x_i, h_i)_{i=1}^{n}$, where $x_i$ is a source segment and $h_i$ its corresponding hypothesis translation, a quality estimation model assigns a score $Q(x_i, h_i)$ to each pair. Filtering can then be formulated as a selection problem:

$$\mathcal{D}_{\text{filtered}} = \{(x_i, h_i) \mid Q(x_i, h_i) \geq \tau\} \tag{2}$$

where $\tau$ is a quality threshold. By retaining only high-quality pairs, data filtering reduces noise in training corpora and leads to more robust and effective MT models.

**RLHF and Reranking.** Quality estimation also plays a central role in Reinforcement Learning from Human Feedback (RLHF) and reranking tasks. In these settings, a system produces a set of candidate translations $h_1, \ldots, h_k$ for a given source $x$. A QE model provides scores $Q(x, h_j)_{j=1}^{k}$, which serve two main purposes: (i) as reward signals for reinforcement learning, where model parameters are updated to maximize expected quality $\mathbb{E}[Q(x, h)]$, and (ii) as reranking criteria, where the final output is chosen by

$$\hat{h} = \arg\max_{h_j} Q(x, h_j) \tag{3}$$

Unlike reference-based metrics, QE directly evaluates $(x, h)$ pairs, making it suitable for iterative optimization or online reranking in production environments. By steering models toward outputs that receive higher QE scores, these methods align system behavior more closely with human preferences without requiring reference translations.

# 3 EXPERIMENTS

## 3.1 EXPERIMENT SETUP

**Dataset.** We use the WMT24++ data (Deutsch et al., 2025) for our experiments. WMT24++ contains the human translation and post-edit data for 55 en→xx language pairs. We focus on a subset of 10 most common languages, including German, Chinese, Spanish, French, Japanese, Korean, Hindi, Arabic, Russian and Portuguese. One important assumption we make is that the post-edit translations from the WMT24++ dataset are error-free, as the translations are verified by human experts and are of high quality. This ensures that any observed trend in changes of metric scores is not attributable to potential errors in the original data.

**Quality Estimation Metrics.** We consider four popular QE metrics spanning two major evaluation frameworks: (i) two regression-based metrics, MetricX-24 QE (Juraska et al., 2024) and CometKiwi (Rei et al., 2022), and (ii) two LLM-as-a-Judge variants, AutoMQM fine-tuned on Gemini-2.0 (Fernandes et al., 2023; Finkelstein et al., 2024) and directly prompted Gemini-2.5-pro model (Gemini Team, 2025), both of which are used to perform MQM-style evaluation.

**Tasks.** To assess and quantify length bias, we evaluate the four proposed QE metrics on two stimulated scenarios introduced in Section 2.2: (i) data filtering and (ii) RLHF and reranking, which correspond to length bias (i) in source texts and (ii) in translation, respectively. For stimulating data filtering, we construct multiple source–translation pairs of equivalent quality and test whether the metrics exhibit systematic preferences based on translation length. For stimulating RLHF and reranking, we compare two translations of equal quality from the same source, partition them into groups by relative length, and examine whether the metrics favor a specific group.

## 3.2 LENGTH BIAS IN SOURCE TEXTS

**Experiment Setup.** To examine length bias in source texts, we construct a dataset from the WMT24++ data as follows: for each language pair, we collect the first 5 segments from each document in the data, denoted as $s_1$, $s_2$, $s_3$, $s_4$ and $s_5$, where each segment is approximately 1-2 sentences long. Then, starting from $s_1$, we gradually append the next segment $s_i$ to the previous

segments to form passages of different lengths. Formally, for each document $d$, we construct a group of 5 passages, denoted as $p_1$, $p_2$, $p_3$, $p_4$ and $p_5$, where $p_i = s_1 + \cdots + s_i$.

We construct the dataset with two considerations. First, selecting segments from the same document keeps the constructed passages within a single domain and minimizes confounds from topic variation. Second, to test our hypothesis, the dataset must include source–translation pairs spanning a wide range of lengths; thus, we concatenate consecutive segments to better preserve inter-segment fluency and reduce boundary-induced disfluency that could potentially depress performance.

Since the metrics inherit different context-window limits from their base models (e.g., 512 tokens for MetricX-24 QE; (Xue et al., 2021)), we standardize the context window to the smallest across metrics, and thus discard documents whose first 5 segments exceed 500 tokens using MetricX-24's tokenizer [1]. We then evaluate each group of 5 passages using distinct metrics and observe the relationship between text length and translation score. To control for score variations between language pairs, we normalize scores within each group by subtracting the score of $p_1$ from $p2$, $p3$, $p4$, and $p5$, and report the score differences instead of raw scores. Figure 2 presents the trend of performance change with line charts over passage index across language pairs from all metrics.

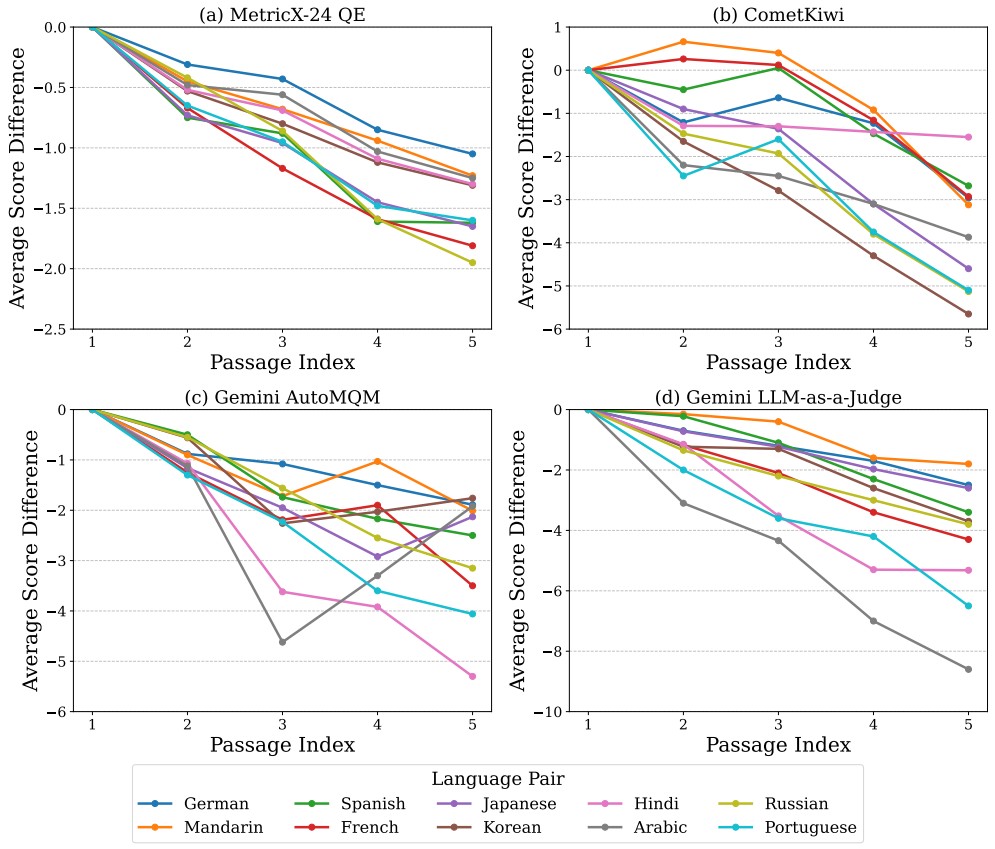

Figure 2: Trends of metric scores with increasing input length across different metrics. Each data-point represents the score change relative to the first passage.

**Results.** As shown in Figure 2, we observe a consistent trend across all language pairs: translation quality deteriorates as passage length increases. Specifically, extending the translation from 1 to 5 segments results in score drops of 1.0–2.0 points for MetricX-24 QE, 1.5–5.6 points for CometKiwi, 1.7–5.3 points for Gemini AutoMQM, and 1.8–8.5 points for Gemini LLM-as-a-Judge. Moreover, Appendix A.1 presents statistics on the proportion of decreasing trends, suggesting a consistent presence of length bias across the majority of cases.

---

[1] google/mt5-xl

To better quantify and understand this behavior, we note that the metrics are trained on MQM-annotated data using the error scheme of Freitag et al. (2021), which assigns penalties of –1 for minor errors and –5 for major errors. Consequently, when translation length increases to five segments, the metrics incorrectly predict an additional minor or major error in these error-free passages. This undesirable sensitivity to length indicates that the metrics exhibit length bias in source texts.

### 3.3 BIAS IN ERROR TYPE AND SEVERITY

The preceding experiments demonstrate that length bias exists in source texts. What remains unclear, however, is whether this bias still appears when the translation contains explicit errors. In addition, we investigate whether the bias is linked to specific combinations of error severity and error type. To this end, we design a follow-up experiment that introduces four controlled categories of errors. Building on the setup in 3.2, we modify the first segment of each passage by inserting one of the following: (a) a major accuracy error, (b) a major fluency error, (c) a minor accuracy error, or (d) a minor fluency error. The formal definitions of these error types, together with the few-shot examples used for prompting, are provided in the Appendix A.3. We then evaluate the passages using MetricX-24 QE and Gemini-based AutoMQM. For simplicity, we report results for English–German and English–Chinese in Figure 3 and Figure 4.

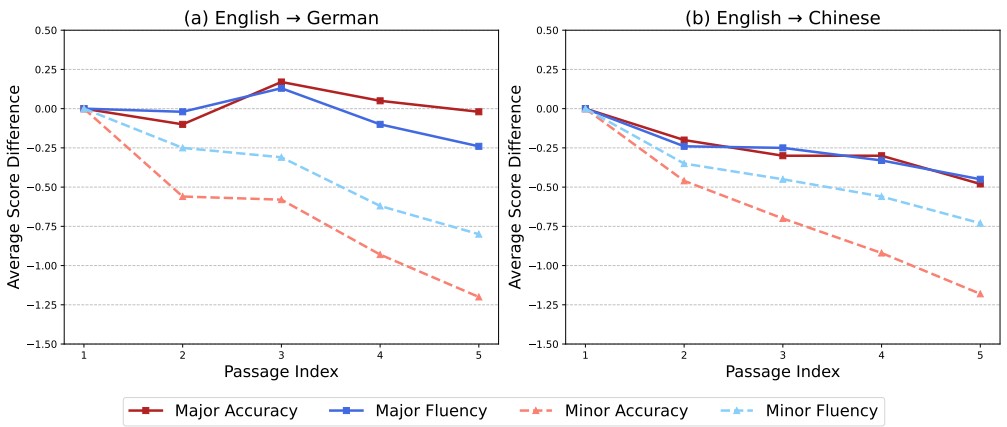

Figure 3: Trends of MetricX-24 QE score with different severity/error types, evaluated on WMT24++ En-De and En-Zh.

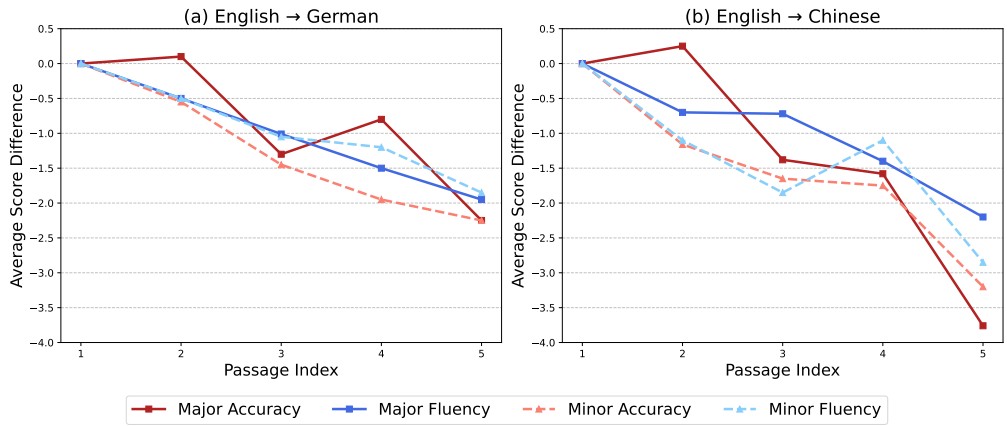

Figure 4: Trends of Gemini AutoMQM score with different severity/error types, evaluated on WMT24++ En-De and En-Zh.

For the regression-based metric, we find that inserting a major error in the first segment tends to reduce the fluctuations in scores as additional error-free segments are appended. In contrast, inserting a minor error has little influence on the existing length bias. This difference can be attributed to the fact that major errors provide a strong signal that outweighs the effect of length bias by delineating a clearer boundary between erroneous and correct sentences, whereas minor errors contribute substantially weaker signals. For the AutoMQM-based metric, conversely, the results suggest no clear bias with respect to error severity. Finally, across both metrics, fluency errors appear to have a smaller impact on score changes than accuracy errors. A plausible explanation is that metrics are generally more sensitive to accuracy errors, while fluency errors are more subjective and tend to show higher levels of disagreement among raters in the training data.

## 3.4 LENGTH BIAS IN TRANSLATION

**Experiment Setup.** To investigate length bias in translation, we sample multiple hypothesis translations from the same source as follows: we first pre-process the source and reference translations from the WMT24++ dataset into chunks that fit within the minimum context window (500 tokens). To ensure meaningful variation while maintaining translation quality, we also discard chunks that are too short (fewer than 200 tokens). Given each source and reference translation, we prompt Gemini-2.5-pro to generate five rephrased translations that preserve the same level of accuracy and fluency with temperature $t = 1$ and top-$p = 0.95$. Then, we retain the two candidates with the minimum and maximum token length, which grants the largest length difference between two candidates of similar quality. We manually verify that the selected candidates remain valid translations.

At this stage, each source chunk is paired with two translations: one shorter and one longer. We compare their lengths and categorize them into the shorter/longer bin. Then, we evaluate both bins with the target metric and compute the proportion of cases in which the shorter translation receives a higher score. To analyze the effect of length differences, we evaluate translation pairs whose lengths differ by increments of 2.5%, ranging from 2.5% to 15%. For each threshold, we filter out pairs that do not meet the required difference. Figure 5 presents the results averaged across all 10 language pairs, for both EN → XX and XX → EN directions using MetricX-24 QE and CometKiwi. The complete tables for individual language pairs are included in Appendix A.2.

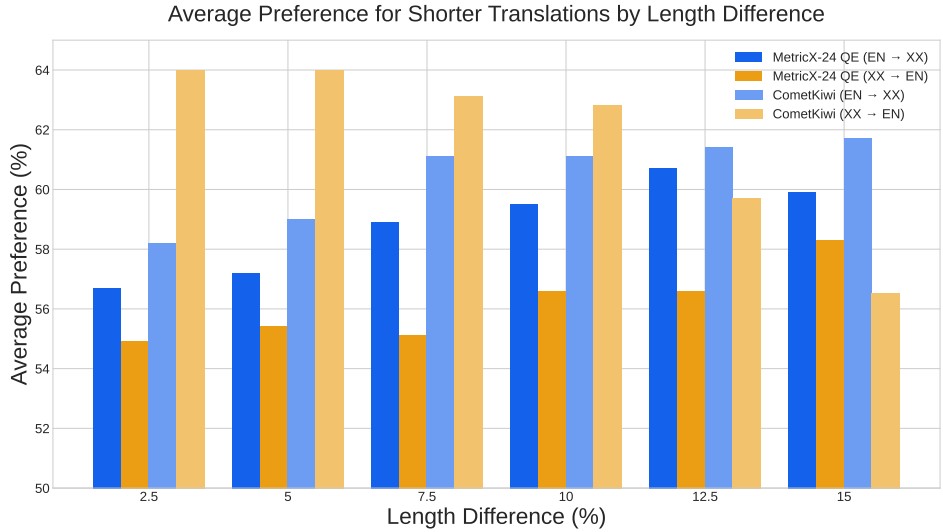

Figure 5: Average preference for shorter translations by length difference.

**Results.** Across all evaluated settings, there is a consistent but modest preference for shorter translations, with average scores ranging from 55% to 64%. This preference becomes more pronounced as the length disparity between the shorter and longer hypotheses increases.

For the EN → XX direction, both MetricX-24 QE and CometKiwi increasingly favor shorter hypotheses. This trend is particularly conspicuous in translations to Spanish, Japanese, and Russian.

Conversely, for the XX → EN direction, the two metrics diverge. MetricX-24 QE exhibits a similar upward trend, most prominently with Spanish, French, Korean, and Russian. In contrast, CometKiwi displays heterogeneous behavior; while it shows an average downward trend (favoring longer hypotheses), this is driven by languages like Korean, Hindi, and Portuguese, while others such as German, Chinese, and Japanese show the opposite upward trend. Complete results for individual language pairs are available in Appendix A.2.

## 4 ANALYSIS

### 4.1 DISTRIBUTION OF MQM SCORES IN WMT DATA

To understand the length bias of models for long-form inputs, we construct Blob-level MQM test sets using WMT data, each consisting of segments up to 500, 1k, or 2k tokens. Figure 6 presents the distribution of gold standard scores and MetricX-24 QE's output scores on these test sets. We notice that (i) the distribution of output scores was skewed, and (ii) as the number of input tokens increases, the number of perfect translations (i.e., gold score = 0) in the source data diminishes, and the output distribution reflects this pattern, with the frequency of zero-score predictions dropping accordingly.

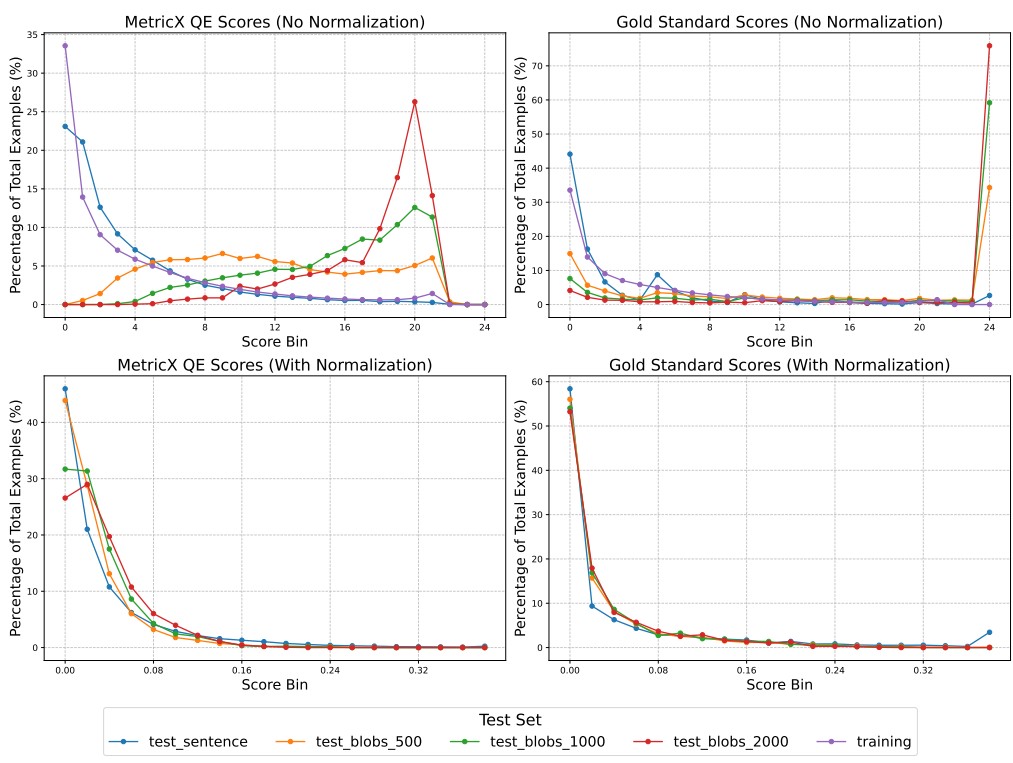

Figure 6: Distribution of MetricX QE outputs and gold standard scores.

### 4.2 MITIGATION: NORMALIZATION AT TRAINING

To address the above length biases, we introduce length normalization, which corrects the skew in the relative frequencies of output scores. With normalization, the model is trained to predict the **error density** of a hypothesis translation, defined as

$$D(x,h) = \frac{R(x,h)}{|h|}$$

where $R(x,h)$ is the error rating of hypothesis $h$ for source $x$ and $|h|$ denotes its segment length. At inference time, we rescale the prediction back to an error rating via

$$\hat{R}(x,h) = D(x,h) \cdot |h|$$

to enable comparison with the unnormalized baseline. Figure 6 shows the gold score distributions with normalization, demonstrating greater consistency across test sets of varying lengths.

To further examine the effectiveness of normalization, we trained a MetricX QE model with the above training-time normalization. Figure 6 shows the distribution of output scores using this trained model. We then replicate the experimental setup in Section 3.2 and report results in Figure 7. To highlight the effect of normalization, we compare the **absolute average score difference** across segment lengths (i.e., the slope of score changes). Results indicate that applying normalization reduces length bias beyond three segments, consistent with our earlier analysis of output score distributions.

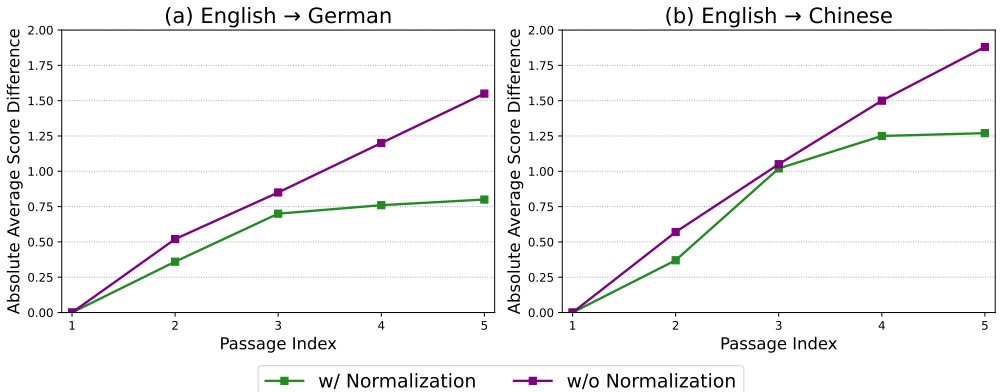

Figure 7: Performance of MetricX-24 QE with and without length normalization. We report the slope of score changes to show that length normalization mitigates the magnitude of length bias.

### 4.3 MITIGATION: REFERENCE-BASED METRICS

When references are available, an alternative solution to mitigate evaluation-time length bias is to incorporate reference-based metrics. Figure 8 compares three classes of MT metrics: reference-based, source-based (QE), and hybrid, under the experiment setup described in Section 3.2. We observe that metrics with access to references exhibit attenuated length bias, plausibly because the reference translation provides an implicit prior on the expected length of a well-formed hypothesis. Nonetheless, these metrics continue to exhibit residual length bias, indicating that reference access alone does not fully eliminate the effect.

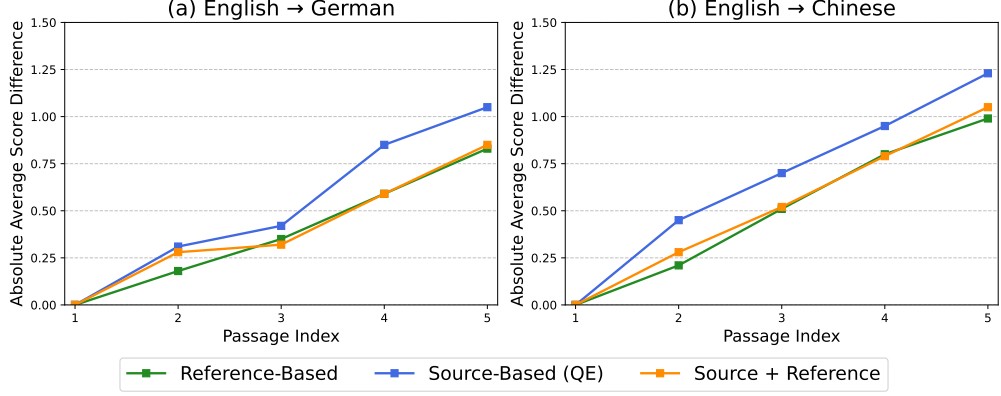

Figure 8: Performance of three classes of MetricX-24: reference-based, source-based (QE), and hybrid. We report the slope of score changes to illustrate that metrics with access to references exhibit reduced length bias, though the effect nonetheless persists.

## 5 RELATED WORK

### 5.1 LENGTH EFFECTS IN MT EVALUATION METRICS

Early reference-based MT metrics explicitly accounted for translation length. For example, BLEU introduced a brevity penalty to discourage overly short translations that can inflate n-gram precision (Papineni et al., 2002). This penalty encourages matching the reference length, addressing that very short translations can yield inflated precision. Similarly, chrF uses an F-score over character n-grams (Popović, 2015), penalizing outputs that are too short or too long via missing or extra content. Nonetheless, these reference-based metrics can still exhibit quirks with length. Papineni et al. (2002) note that neither the brevity penalty nor n-gram precision alone perfectly enforces appropriate length. Moreover, corpus-level BLEU can paradoxically rise on grouped longer texts due to more matching opportunities (Peng et al., 2025), even though longer texts are generally more challenging. These observations highlight the importance of handling length carefully in evaluation metrics to avoid misleading results.

Beyond reference-based metrics, learned neural metrics and LLM-based evaluators also struggle with length effects. Recent analyses show that metrics like COMET can assign surprisingly decent scores to empty or partial outputs if not trained on such edge cases (Rei et al., 2020). Conversely, some metrics are biased toward rewarding excessively long outputs: Domhan & Zhu (2025) show that state-of-the-art LLMs lack length invariance when assessing document-level translations and generally detect fewer error spans compared to segment-level evaluation. These results indicate that ensuring evaluation metrics remain fair and length-robust is still a concern. Hence, our work situates itself in this context by examining whether current QE methods penalize translations simply for being longer, rather than truly worse.

### 5.2 BIASES IN QUALITY ESTIMATION MODELS

Modern QE models have shifted from feature-based frameworks such as QuEst (Specia et al., 2013) to neural architectures, achieving strong results on recent benchmarks (Rei et al., 2022; 2023), with representative examples including OpenKiwi, TransQuest, and CometKiwi (Kepler et al., 2019; Ranasinghe et al., 2020; Rei et al., 2022). Despite these improvements, researchers have noted that learned QE models can pick up spurious correlations or biases from their training data. For instance, Behnke et al. (2022) identifies a partial input bias in state-of-the-art QE: the model over-relies on target-side fluency features that correlate with high quality in the training data rather than truly evaluating adequacy, causing the model to give high scores to outputs that are fluent and grammatically correct, even if they fail to preserve the source meaning.

Such biases in QE often stem from limitations in training data. QE models are typically supervised on human ratings of MT outputs, such as WMT Direct Assessments or MQM annotations. If these datasets are skewed or unrepresentative, the QE model will reflect that skew. For instance, recent work observed that automatic metrics often score human-generated translations lower than one would expect, compared to MT outputs, implying a bias against human style (Deutsch et al., 2025). This likely arises because the metrics are trained predominantly on machine outputs and have not learned to handle the nuances of human translations, which may use more diverse or free wording. Similarly, domain and language biases have been reported: metrics can overfit to score distributions of particular domains or language pairs (Zouhar et al., 2024), and may not generalize well to others without adjustment. Complementary to these findings, we probe whether the observed length bias in QE correlates with the under-representation of long-form, error-free examples in the training data.

## 6 CONCLUSION

In this work, we show that contemporary QE models lack length invariance when assessing long-form translations: they inflate error predictions as translations grow longer and exhibit a systematic bias toward shorter candidates when evaluating alternatives from the same source. We trace this bias to training distributions that under-represent long, clean examples, and we propose two remedies: (i) a training-time length-normalization objective and (ii) an evaluation-time combination with reference-based metrics when references are available. Together, these interventions reduce length bias and move QE toward more robust, length-agnostic assessment.

## 7 STATEMENTS

### 7.1 ETHICS STATEMENT

We do not foresee any ethical concerns or potential risks arising from this work. All evaluation metrics and datasets are either open-sourced or accessible through public APIs, as described in Section 3.1. The large language models (LLMs) used in our experiments are also publicly available. Within our research context, the outputs generated (i.e., numerical scores from metrics and error ratings from LLMs) do not contain or propagate harmful, unsafe, or sensitive information. Accordingly, we believe that this study fully adheres to the Code of Ethics.

### 7.2 REPRODUCIBILITY STATEMENT

We have taken several steps to ensure the reproducibility of our work. All evaluation metrics and datasets used in our experiments are open-sourced or publicly accessible. The procedures for constructing the experimental data are described in detail in Sections 3.2 and 3.4, with additional clarifications provided in the appendix. Upon acceptance, we will release the scripts necessary to replicate all experiments, including data processing, model evaluation, and result reporting. Together, these resources are intended to enable independent verification and extension of our findings.

### 7.3 LLM USAGE DISCLOSURE

The authors acknowledge the use of the Large Language Model Gemini 2.5 Pro for the purpose of language editing and polishing. The model was used to refine grammar, syntax, and phrasing to enhance the readability of the manuscript. The research, experimental design, and conclusions presented herein are the original work of the authors.

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

# A APPENDIX

## A.1 PROPORTION OF SCORES WITH DECREASING TRENDS IN DATA FILTERING

Additionally, we analyze the proportion of documents exhibiting a decreasing trend in translation scores. For each language pair, we compare the score of passage 1 (containing segment 1) with passage 5 (containing segments 1–5), and report (i) the total number of documents and (ii) the proportion that display a decreasing trend. The results for MetricX-24 QE are summarized in Table 1. Overall, about 80% of the documents across 10 languages show a decreasing trend, indicating that length bias is pervasive in most cases.

| Language | # of Documents | Proportion of scores with decreasing trends (%) |
|---|---|---|
| Aggregate | 472 | 80.1 |
| German (de_DE) | 48 | 85.4 |
| Chinese (zh_CN) | 53 | 79.2 |
| Spanish (es_MX) | 47 | 74.5 |
| French (fr_FR) | 45 | 82.2 |
| Japanese (ja_JP) | 54 | 90.7 |
| Korean (ko_KR) | 46 | 82.6 |
| Hindi (hi_IN) | 37 | 75.7 |
| Arabic (ar_EG) | 48 | 68.8 |
| Russian (ru_RU) | 48 | 79.2 |
| Portuguese (pt_BR) | 46 | 80.4 |

Table 1: Proportion of scores with decreasing trends with MetricX-24 QE

## A.2 NUMERICAL RESULTS FOR THE RLHF AND RERANKING EXPERIMENTS

Tables 2, 3, 4, 5 present the numerical results for the RLHF and Reranking experiments in Section 3.4 in the tables below.

| Target Language | Preference for Shorter Translations (%) by Length Difference | | | | | |
|---|---|---|---|---|---|---|
| | 2.5% | 5% | 7.5% | 10% | 12.5% | 15% |
| German (de_DE) | 55.4 (101) | 53.8 (93) | 55.4 (74) | 58.3 (48) | 48.6 (35) | 45.8 (24) |
| Chinese (zh_CN) | 52.3 (86) | 54.4 (79) | 53.1 (64) | 53.1 (49) | 58.3 (36) | 56.0 (25) |
| Spanish (es_MX) | 49.0 (104) | 51.2 (80) | 59.1 (44) | 60.9 (23) | 66.7 (15) | 63.6 (11) |
| French (fr_FR) | 68.9 (106) | 68.0 (97) | 68.4 (76) | 70.6 (51) | 68.8 (32) | 70.0 (20) |
| Japanese (ja_JP) | 63.0 (81) | 63.2 (76) | 66.7 (54) | 67.5 (40) | 72.4 (29) | 75.0 (20) |
| Korean (ko_KR) | 54.3 (105) | 55.1 (98) | 56.8 (74) | 48.0 (50) | 45.5 (33) | 50.0 (24) |
| Hindi (hi_IN) | 44.8 (116) | 45.5 (112) | 47.5 (101) | 48.3 (87) | 48.7 (78) | 44.8 (67) |
| Arabic (ar_EG) | 70.9 (103) | 72.0 (100) | 71.1 (97) | 70.8 (89) | 69.1 (68) | 67.9 (53) |
| Russian (ru_RU) | 61.8 (102) | 61.1 (95) | 61.1 (72) | 67.3 (55) | 66.7 (36) | 64.3 (28) |
| Portuguese (pt_PT) | 46.2 (104) | 47.4 (95) | 49.3 (69) | 50.0 (42) | 61.9 (21) | 61.1 (18) |

Table 2: Results on MetricX-24 QE's translation length preferences for EN → XX language pairs

## A.3 ERROR TYPE / SEVERITY PERTURBATION PROMPT

Figures 9, 10, 11, 12, 13 present the prompts used for error perturbation with different types and severity using Gemini-2.5-pro.

## A.4 LLM-AS-A-JUDGE PROMPT

Figure 14 presents the prompts used by LLM-as-a-Judge with Gemini-2.5-pro.

| Source Language | Preference for Shorter Translations (%) by Length Difference | | | | | |
| --- | --- | --- | --- | --- | --- | --- |
| | 2.5% | 5% | 7.5% | 10% | 12.5% | 15% |
| German (de_DE) | 45.5 (101) | 45.2 (93) | 42.9 (63) | 43.8 (48) | 36.4 (33) | 40.0 (25) |
| Chinese (zh_CN) | 36.1 (83) | 35.4 (82) | 34.2 (73) | 32.8 (61) | 30.8 (52) | 31.7 (41) |
| Spanish (es_MX) | 63.1 (103) | 62.6 (91) | 62.9 (70) | 68.4 (57) | 62.2 (45) | 65.8 (38) |
| French (fr_FR) | 42.1 (107) | 43.4 (99) | 46.2 (78) | 52.6 (57) | 52.9 (34) | 52.0 (25) |
| Japanese (ja_JP) | 54.2 (83) | 52.5 (80) | 51.5 (68) | 51.9 (52) | 54.3 (35) | 52.0 (25) |
| Korean (ko_KR) | 57.1 (105) | 60.2 (98) | 60.2 (83) | 59.7 (67) | 61.8 (55) | 65.2 (46) |
| Hindi (hi_IN) | 75.9 (116) | 75.7 (115) | 77.5 (111) | 78.5 (107) | 77.9 (104) | 78.4 (102) |
| Arabic (ar_EG) | 59.2 (103) | 61.2 (98) | 61.7 (94) | 65.5 (87) | 68.8 (80) | 70.1 (67) |
| Russian (ru_RU) | 56.9 (102) | 58.6 (99) | 60.5 (86) | 60.9 (69) | 67.9 (56) | 69.6 (46) |
| Portuguese (pt_PT) | 59.0 (105) | 58.8 (97) | 53.2 (77) | 51.6 (62) | 53.1 (49) | 58.5 (41) |

Table 3: Results on MetricX-24 QE's translation length preferences for XX → EN language pairs

| Target Language | Preference for Shorter Translations (%) by Length Difference | | | | | |
| --- | --- | --- | --- | --- | --- | --- |
| | 2.5% | 5% | 7.5% | 10% | 12.5% | 15% |
| German (de_DE) | 61.4 (101) | 62.4 (93) | 63.5 (74) | 64.6 (48) | 68.6 (35) | 75.0 (24) |
| Chinese (zh_CN) | 60.5 (86) | 59.5 (79) | 59.4 (64) | 57.1 (49) | 52.8 (36) | 52.0 (25) |
| Spanish (es_MX) | 53.8 (104) | 58.8 (80) | 68.2 (44) | 65.2 (23) | 66.7 (15) | 63.6 (11) |
| French (fr_FR) | 53.8 (106) | 54.6 (97) | 61.8 (76) | 64.7 (51) | 68.8 (32) | 70.0 (20) |
| Japanese (ja_JP) | 64.2 (81) | 64.5 (76) | 72.2 (54) | 67.5 (40) | 65.5 (29) | 65.0 (20) |
| Korean (ko_KR) | 57.1 (105) | 57.1 (98) | 54.1 (74) | 62.0 (50) | 69.7 (33) | 66.7 (24) |
| Hindi (hi_IN) | 60.3 (116) | 61.6 (112) | 60.4 (101) | 58.6 (87) | 61.5 (78) | 59.7 (67) |
| Arabic (ar_EG) | 49.5 (103) | 49.0 (100) | 50.5 (97) | 50.6 (89) | 48.5 (68) | 49.1 (53) |
| Russian (ru_RU) | 61.8 (102) | 62.1 (95) | 62.5 (72) | 63.6 (55) | 63.9 (36) | 60.7 (28) |
| Portuguese (pt_PT) | 59.6 (104) | 60.0 (95) | 58.0 (69) | 57.1 (42) | 47.6 (21) | 55.6 (18) |

Table 4: Results on CometKiwi's translation length preferences for EN → XX language pairs

| Source Language | Preference for Shorter Translations (%) by Length Difference | | | | | |
| --- | --- | --- | --- | --- | --- | --- |
| | 2.5% | 5% | 7.5% | 10% | 12.5% | 15% |
| German (de_DE) | 68.3 (101) | 69.9 (93) | 77.8 (63) | 75.0 (48) | 69.7 (33) | 64.0 (25) |
| Chinese (zh_CN) | 74.7 (83) | 74.4 (82) | 72.6 (73) | 75.4 (61) | 75.0 (52) | 75.6 (41) |
| Spanish (es_MX) | 58.3 (103) | 61.5 (91) | 61.4 (70) | 56.1 (57) | 55.6 (45) | 50.0 (38) |
| French (fr_FR) | 73.8 (107) | 72.7 (99) | 70.5 (78) | 75.4 (57) | 70.6 (34) | 60.0 (25) |
| Japanese (ja_JP) | 77.1 (83) | 77.5 (80) | 77.9 (68) | 76.9 (52) | 77.1 (35) | 80.0 (25) |
| Korean (ko_KR) | 61.9 (105) | 59.2 (98) | 57.8 (83) | 52.2 (67) | 43.6 (55) | 39.1 (46) |
| Hindi (hi_IN) | 42.2 (116) | 41.7 (115) | 39.6 (111) | 37.4 (107) | 35.6 (104) | 35.3 (102) |
| Arabic (ar_EG) | 57.3 (103) | 56.1 (98) | 55.3 (94) | 55.2 (87) | 55.0 (80) | 52.2 (67) |
| Russian (ru_RU) | 67.6 (102) | 66.7 (99) | 65.1 (86) | 71.0 (69) | 67.9 (56) | 67.4 (46) |
| Portuguese (pt_PT) | 59.0 (105) | 59.8 (97) | 53.2 (77) | 53.2 (62) | 46.9 (49) | 41.5 (41) |

Table 5: Results on CometKiwi's translation length preferences for XX → EN language pairs

```
You are an expert in [[query_language_name]] and [[translation_language_name]].
You are presented with the following [[query_language_name]] source text and its
[[translation_language_name]] translation.

[[task_description]]

Source text: [[query]]

Translation: [[translation]]

IMPORTANT NOTE: Only insert one such error in the translation. Do not insert any
other error. Do not significantly increase or decrease the length of the
translation. Do not bold any words or number the lines.
```

Figure 9: Error perturbation prompt

```
Your task is to revise the provided translation to introduce one major accuracy error.
This error should be a significant mistranslation, omission, or addition that alters a
key piece of information from the source text. The error must be noticeable and
disrupt the intended meaning, but the sentence should remain grammatically correct and
generally understandable.

For example (the error is labeled with <v></v>):

Source text: これでも一番いいオファーだったわけです それくらい球界内で注目されていなかったアクーニャ
当時は体が小さくそれがスカウト内で 野球選手としての将来に対する疑問に 繋がったのかもしれませんが プロ入
りしてからアクーニャは瞬く間に 球界ナンバーワンクラスの有望壁へと成長し 2018年のMLBデビュー以降も オー
ルスターレベルの活躍を続けているんですから ブレーブスは本当にいい選手を発掘しましたよね

Translation with major accuracy error: これ已经是当时最好的报价了。<v>这足以说明阿库尼亚在第9轮之
前并没有受到太多关注。</v>当时他的体型较小，这可能导致球探对他作为棒球运动员的未来产生了质疑。但是，在
成为职业选手后，阿库尼亚迅速成长为第9轮中最有前途的球员之一。自2018年在MLB首秀以来，他一直保持着全明星
级别的表现。勇士队确实在这位球员身上下了一个很好的赌注，不是吗？
```

Figure 10: Major accuracy prompt

```
Your task is to revise the provided translation to introduce one major fluency error.
This error should be a significant grammatical, spelling, or punctuation mistake that
makes the translated text sound unnatural and difficult to read.
The error must disrupt the flow of the text, but a reader should still be able to
understand the intended message.

For example (the error is labeled with <v></v>):

Source text: やってみよう パトロール戻りました そいっす いやーそれにしても暇だな それだけ平和ってこ
とだよな そういえば二人とも昔から警官になりたかった？ いや俺はプロ野球選手になりたかった マジ?俺も って
ことは森本も？

Translation with major fluency error: <v>让我们试试吧 巡逻回来了 是的 哎呀</v>，话说回来真是闲
啊 这不就意味着很和平吗 说起来，你们两个从小就想当警察吗？ 不，我本来想成为职业棒球选手 真的吗？ 我也是
那么森本也是吗？
```

Figure 11: Major fluency prompt

Your task is to revise the provided translation to introduce one minor accuracy error. This error should be a subtle mistranslation (such as using a word with a slightly incorrect meaning) or a minor omission that is technically incorrect. The error must not significantly change the overall meaning of the sentence or hinder a reader's comprehension.

For example (the error is labeled with <v></v>):

Source text: 島リスのジャンプシーンを狙うなら こういうところから撮るんですが 始めての方はまず餌場から撮りましょう 食事中と休憩中の島リスは撮りどころです じっとしている島リスがいたらすぐに写真撮影をして まだ撮れそうだったら動画に切り替えて撮影をします。

Translation with minor accuracy error: 如果想拍摄松鼠跳跃的场景，应该从这样的地方拍摄。但是初学者最好先从<v>喂食区</v>开始拍摄。正在进食和休息的松鼠是很好的拍摄对象。如果看到松鼠静止不动，就立即拍照。如果感觉还有机会继续拍摄，就切换到视频模式进行拍摄。

Figure 12: Minor accuracy prompt

Your task is to revise the provided translation to introduce one minor fluency error. This error should be a small grammatical, spelling, or punctuation mistake. The error must be subtle and should not disrupt the natural flow of the translation or prevent a reader from easily understanding it.

For example (the error is labeled with <v></v>):

Source text: 企業規模が小さくなるにつれ、策定率が下がる傾向にある。小さな企業ほど人的、時間的に余裕がないほか、具体的にどうすればよいか分からないという話も聞く。策定に向けて企業が求める支援や取り組みとして、災害対応力向上やＢＣＰ策定のセミナー実施を求める声は多く、ＢＣＰの理解を広げることから始めたい。そのための行政支援も必要だろう。

Translation with minor fluency error: 随着企业规模的缩小，制定率呈现下降趋势。小企业的人力和时间都比较紧缺，而且我也听说他们不知道具体该怎么做。在制定BCP的过程中，企业所需要的支持和行动是，有很多人希望<v>通过开展应对灾害能力提升和BCP制定研讨会来寻求支持</v>，所以我想首先从扩大对BCP的理解开始。也需要行政上的支持。

Figure 13: Minor fluency prompt

```
You are an annotator for the quality of machine translation. Your task is to
identify errors and assess the quality of the translation. Based on the source
segment and machine translation surrounded with triple backticks, identify
error types in the translation and classify them. The categories of errors
are: accuracy (addition, mistranslation, omission, untranslated text), fluency
(character encoding, grammar, inconsistency, punctuation, register, spelling),
style (awkward), terminology (inappropriate for context, inconsistent use),
non-translation, other, or no-error. Each error is classified as one of three
severities: critical, major, and minor. Critical errors inhibit comprehension
of the text. Major errors disrupt the flow, but what the text is trying to say
is still understandable. Minor errors are technically errors, but do not
disrupt the flow or hinder comprehension. Make sure your response is a strict
and valid json object that could be parsed with json.loads() in python.

Example #1 (major error):
Source:
```参加者はくじ引きで対局相手を決定し「よろしくお願いします」のあいさつと振り駒で対局開始。県
予選は持ち時間がリーグ15分、トーナメント20分いずれも秒読み30秒で戦った。交流戦は６歳−小６が
スイス式トーナメントで戦った。```
Translation:
```参加者通过抽签决定比赛对手，"请多关照"的问候和"摇旗"开始比赛。县预选赛在联赛中每局15分钟,
锦标赛每局20秒钟，每局30秒的秒读。交流赛6岁-小6使用瑞士式锦标赛。```
Output:
```[{"span": "摇旗", "severity": "major", "category": "Accuracy/
Mistranslation"}]```

Example #2 (minor error):
Source:
```Today we're making some beautiful vintage Easter craft. Keep watching! I'm
Brandy and this is Making it my own DIY. Welcome! The first project are going
to be two Easter ornaments. I'm going to start off with two little ornaments
here and you can use whatever you have on hand. I've got these little decor
eggs from Dollar Tree.```
Translation:
```Heute machen wir ein paar wunderschöne Vintage-Ostereier-Bastelarbeiten.
Bleib dran! Ich bin Brandy und das ist mein eigenes DIY. Willkommen! Das erste
Projekt sind zwei Osterdekorationen. Ich fange mit zwei kleinen Dekoeiern an,
und du kannst alles verwenden, was du zur Hand hast. Ich habe diese kleinen
Dekoeier vom Dollar Tree.```
Output:
```[{"span": "Bastelarbeiten", "severity": "minor", "category": "Style/
Unnatural or awkward"}]```

Example #3 (no error):
```
Source:
```Our position is clear, the trade association Brewers of Europe wrote to
MEPs and government negotiators (10 January). "If targets and mandatory
requirements in the PPWR apply to beer then there is a legal obligation that
they apply to all alcoholic beverage categories."```
Translation:
```Nuestra posición es clara, escribió la asociación comercial Brewers of
Europe a los eurodiputados y negociadores gubernamentales (10 de enero). "Si
los objetivos y los requisitos obligatorios en la PPWR se aplican a la
cerveza, entonces existe una obligación legal de que se apliquen a todas las
categorías de bebidas alcohólicas".```
Output:
```[]```

[[src_lang]] Source:
```[[source]]```
[[tgt_lang]] Translation:
```[[translation]]```
Output:
```

Figure 14: LLM-as-a-Judge Prompt

