# OpenReview forum: "Penalizing Length: Uncovering Systematic Bias in Quality Estimation Metrics"
_ICLR.cc/2026/Conference — Submitted to ICLR 2026_

### Official Review · Reviewer_wgfj · 2025-10-29

**Soundness:** 2
**Presentation:** 2
**Contribution:** 2
**Rating:** 2
**Confidence:** 4

**Summary:**

In this work, the authors address the issue of length bias in quality estimation (QE) metrics, where such metrics tend to favor shorter translations and penalize correct but longer ones.

To mitigate this problem, the authors propose two approaches:

(i) normalizing gold scores by translation length during metric training, and

(ii) incorporating reference translations when they are available.

To analyze length bias, the authors conduct two complementary experiments. First, they artificially increase translation length by concatenating correct translations across multiple segments and observe how QE metric scores change. Second, they generate translations of varying lengths for the same source sentence—by prompting a large language model to maintain accuracy and fluency similar to a reference translation—and then compare metric scores across different length buckets.

Overall, the results demonstrate that the proposed methods can effectively reduce length bias to some extent.

**Strengths:**

•	The paper addresses an important issue in QE metrics, as length bias can affect decision-making in downstream applications.

•	Several thoughtful analyses are presented to highlight and motivate the problem of length bias in QE metrics.

•	The use of multiple QE metrics helps demonstrate the generalizability of the issue.

•	The proposed solution of training QE metrics on length-normalized scores is simple and intuitive.

**Weaknesses:**

•	No meta-evaluation is performed to validate the effectiveness of the proposed solution, making it difficult to assess its reliability or whether it introduces unintended artifacts. The authors could consider reviewing recent work on meta-evaluation, e.g., https://www2.statmt.org/wmt24/pdf/2024.wmt-1.35.pdf and https://www2.statmt.org/wmt24/pdf/2024.wmt-1.2.pdf

•	The analyses assume that translations are error-free, which is not always the case. According to the WMT findings paper (https://www2.statmt.org/wmt24/pdf/2024.wmt-1.1.pdf)), reference translations are not always preferred, so some observed bias could be due to errors in the translations themselves. A small-scale human evaluation could help validate this aspect. For example: collecting MQM or DA scores for each individual segment as well as concatenated segments and check whether the length bias is also present from human perspective.

•	The proposed approach of training metrics with normalized scores lacks sufficient training details. Providing information on the training data and hyperparameters would improve reproducibility.

•	It is unclear why the second proposed solution—using reference translations to mitigate length bias—is considered a contribution. This approach does not address the issue in reference-free QE models, which are the main focus of the paper.

**Questions:**

N/A

---

> ### Author Response · Authors · 2025-11-25
>
> Thank you for your valuable and constructive feedback. We address your questions below.
>
>
> **W1: Meta Evaluation**
>
> Thank you for noting this. We want to first clarify that our work focuses on analyzing the presence and causes of length bias in QE metrics rather than proposing a new evaluation metric. As a result, we view length normalization as a diagnostic tool for understanding the bias rather than a fully developed replacement metric.
>
> Regarding human evaluation, the WMT24++ dataset used in our experiments already includes human evaluation results. As stated in Section 2.2 of the WMT24++ paper (https://arxiv.org/pdf/2502.12404), the reference translations and post-edits were produced by professional translators. Since we select segments labeled as error-free for our experiments, the corresponding human judgments for these segments are naturally error-free as well.
>
> Furthermore, although the translations and post-edits were performed at the segment (paragraph) level, the translators were provided with the corresponding document context to aid in their understanding of the English text. Therefore, concatenating these segments into longer paragraphs is unlikely to introduce cross-segment fluency errors. Under these assumptions, we show that the examined QE metrics incorrectly predict increasing numbers of errors as the input length grows, even though human evaluation indicates no change in translation quality. We therefore conclude that this discrepancy reveals that QE metrics exhibit length bias. We will add this clarification to the dataset introduction section in the revised draft.
>
>
> **W2: Data Validity**
>
> We would like to clarify that the dataset used in our experiments is WMT24++ (https://arxiv.org/pdf/2502.12404) rather than WMT24. In addition to the original WMT24 data, WMT24++ includes reference translations and post-edits produced by professional translators. Moreover, as noted above, the translators were provided with document-level context to support their understanding of the source text. From a human‐evaluation perspective, this represents the highest translation quality available in the dataset, and we therefore treat these translations as error-free.
>
>
> **W3: Training Details**
>
> Thank you for the suggestion. We provide additional training details for the length-normalized MetricX model below.
>
> ***Training data.*** We follow the standard data setup used in prior work on MetricX. Specifically, we use the WMT20–22 Metrics Shared Task data (https://www.statmt.org/wmt22/metrics/index.html; https://aclanthology.org/2022.wmt-1.2.pdf) as the training set, which includes human-scored translation pairs across multiple language directions and quality levels. For model selection and validation, we use the WMT23 data (https://www2.statmt.org/wmt23/) as the development set. This setup ensures that the model is trained on diverse quality judgments while being evaluated on a more recent distribution.
>
> ***Training hyperparameters.*** The model is fine-tuned using MetricX as the base model for 10,000 steps with a batch size of 256. The learning rate starts at 2e-5 and decays to 1e-6 using cosine decay, following a 100-step linear warmup to stabilize early training. We adopt the same optimization settings and preprocessing pipeline as MetricX to ensure comparability, with the only modification being the inclusion of our length-normalization objective.
>
>
> **W4: Reference-based model**
>
> Thank you for the insightful suggestion. Our original motivation for including the analysis using reference translations to mitigate length bias was the following: we initially investigate why length bias arises and hypothesize that QE models exhibit this bias because, unlike reference-based models, they do not know what the reference translation length should be. As a result, they develop a bias regarding the expected length of a “good” solution. Comparing QE models with reference-based and hybrid models helps us (1) confirm that the length bias is indeed more pronounced in QE models, and (2) understand how far QE models are from reference-based models in terms of length sensitivity.
>
> However, we now agree that this analysis is better suited as supplementary material rather than a core contribution. Since our primary goal is to reveal the presence of length bias in QE models and to propose a mitigation strategy specific to QE models, we will move this section to the Appendix and update the Abstract and Introduction sections accordingly in the revised draft.

---

> > ### Comment · Reviewer_wgfj · 2025-11-27
> >
> > Thank you to the authors for carefully reading my concerns and providing clarifications. Reading the abstract initially gave me the impression that the contributions of the paper—(i) analyzing length bias and (ii) proposing a solution—were each intended to constitute roughly 50% of the work. Now that you have clarified that the primary focus is on analysis, I have updated my scoring perspective accordingly. However, I still feel that the analysis might be a better fit for a short (4–5 page) paper.
> >
> > Overall, in my humble opinion, it would be beneficial to keep the paper’s focus entirely on the analysis and not include the mitigation strategy based on applying length normalization during training. Since there is no meta-evaluation for this new metric (which I believe could itself be a strong contribution), it remains unclear whether the community should adopt it. You might consider preparing a separate paper dedicated solely to addressing the length bias issue in QE metrics.

---

### Official Review · Reviewer_5fA6 · 2025-10-30

**Soundness:** 4
**Presentation:** 3
**Contribution:** 3
**Rating:** 8
**Confidence:** 5

**Summary:**

the paper explores lenght bias in quality estimation
it shows that longer segments are penalised by quality current estimation metrics, both autoregressive as well as LLM-based

two possibilities for reducing the effect are proposed: normalisation of scores over the segment length, and using reference translations as additional input

**Strengths:**

the paper describes an interesting observation regarding evaluating long segments without references

the analysis is sound and the findings are useful

the paper is clearly written

**Weaknesses:**

no major weaknesses

a few small details could be improved (see Questions)

**Questions:**

Related work should be placed after introduction, not at the end

what are tokens exactly? sub-word units?



3.2  the described MQM score is not normalised over the text length (one major error brings -5 points, no matter whether it is one error in a segment of 10 words or one error in a segment of 100 words) -- could it have some influence to the observed length bias?

it seems that this is precisely the idea for mitigation described in 4.2 => it could be discussed clearly already in 3.2

---

> ### Author Response · Authors · 2025-11-25
>
> Thank you for your valuable and detailed feedback, as well as for your kind appreciation of our paper. We address your questions below.
>
> **Q1: Token**
>
> Thank you for the question. By tokens, we refer to the subword units used by the underlying tokenizer of the MetricX QE models, which are based on the T5 architecture. All token counts reported in the paper are measured using the same tokenizer employed by T5/MetricX.
>
> **Q2: MQM score**
>
> Thank you for raising this point. The MQM scoring scheme indeed applies fixed penalties that are not normalized by text length. However, in our experiments, the segments we concatenate are all labeled as error-free by human translators, so their true MQM score remains 0 regardless of length. Since no human-judged errors are introduced as the text becomes longer, any increase in predicted errors by the QE models cannot be attributed to MQM’s length-insensitive weighting, but rather to the models’ own length bias.
>
> Additionally, our analysis in Section 4.1 suggests that the observed length bias likely originates from skewed distributions of output scores in the models’ training data. Section 4.2 then further shows that length normalization mitigates this issue by correcting the skew in the relative frequencies of output scores. We will clarify this in the revised draft.
>
> **Q3: Writing**
>
> Thank you for your suggestions regarding the section organization. We will update the next draft accordingly.

---

### Official Review · Reviewer_wBLz · 2025-10-31

**Soundness:** 3
**Presentation:** 3
**Contribution:** 2
**Rating:** 4
**Confidence:** 4

**Summary:**

Interesting research for MT QE.
-  The study addresses an underexplored yet critical issue—length bias in QE metrics, which are vital for reference-free MT evaluation and RL tasks. By focusing on a practical vulnerability ignored by prior research, it offers meaningful insights into metric reliability, making the research highly engaging and relevant.
-  It systematically uncovers the "length penalty" bias: QE metrics spuriously overpredict errors for longer but error-free translations, verified via controlled experiments of concatenating high-quality segments, ensuring rigorous and convincing findings.
-  The research identifies another distinct bias—QE metrics disproportionately favor shorter candidates when multiple translations of the same source exist. Complemented by effective mitigation strategies (length normalization, reference supplementation), it provides actionable solutions for more equitable evaluations.

**Strengths:**

Good research problems.
- The research targets an underexplored yet practically consequential issue—length bias in QE metrics—filling a critical gap in existing literature and addressing a vulnerability that misguides downstream MT applications like reranking and reinforcement learning.
- It adopts a systematic and rigorous experimental design, with two complementary experiments (concatenating error-free segments, comparing multi-candidate translations) across 10 language pairs, ensuring reliable and generalizable findings.
- The study not only uncovers length biases but also proposes two actionable mitigation strategies (length normalization in training, reference supplementation in evaluation) with validated effectiveness, providing direct value for improving QE metric robustness.

**Weaknesses:**

Not deep analysis in this research.
- It lacks error analysis of high-quality translations; the study uses "error-free" segments but fails to elaborate on error types/distributions that might interact with length, limiting insights into how bias interacts with actual translation quality.
- There is no analysis of the impact of evaluation model size—neither regression-based models nor LLM-as-a-Judge approaches are examined for whether model scale correlates with the severity of length bias.
- The deep-rooted causes of length penalty remain unexplored; the study identifies the bias but does not investigate why QE metrics conflate length with quality (e.g., training data biases, model architectural limitations).
- It neglects differential analysis across QE metric types: the research groups regression-based and LLM-as-a-Judge models together, offering no insights into whether one category exhibits stronger length bias or responds better to mitigation strategies.

**Questions:**

- Differerent translation errors analysis will be helpful.
- Different model and model size analysis will be helpful.

---

> ### Author Response · Authors · 2025-11-25
>
> Thank you for your valuable and constructive feedback. We address your questions below.
>
> **W1: Error Analysis**
>
> Thank you for the suggestions. We agree that examining how length bias interacts with translation quality is important. Section 3.3 of our paper includes an analysis of how different error types and severities influence the magnitude of length bias. For example, we observe that regression-based metrics show greater score fluctuations when a minor error appears early, while major errors have little effect, and AutoMQM-based metrics do not exhibit a clear pattern with respect to error severity. We also find that fluency errors have a smaller impact on score changes than accuracy errors, likely because metrics are more sensitive to accuracy issues and fluency judgments tend to be more subjective.
>
> We would be grateful if you could let us know whether this level of analysis addresses your concern, or if you would prefer a finer-grained breakdown of error types (e.g., decomposing accuracy errors into syntactic errors, semantic errors, omissions, etc.).
>
> **W2: Model Size Analysis**
>
> Thank you for pointing this out. Some of the models we examined, such as CometKiwi and Gemini LLM-as-a-Judge, are only available in a single model size. However, MetricX-24 provides multiple variants (Large, XL, and XXL; our paper uses the full-precision float32 XL variant). We present a comparison of these MetricX-24 model sizes below.
>
> Note: Due to computational constraints, we use the bfloat16 variant of the XXL model (https://huggingface.co/google/metricx-24-hybrid-xxl-v2p6-bfloat16), and for a fair comparison, we also evaluate the bfloat16 variants of the Large and XL models.
>
> ***Experiment #1***: Bias in Source Text Length
>
> We repeat the experiment from Section 3.2, which evaluates bias with respect to source text length, using all three MetricX-24 variants. We observe a clear relationship between model size and the severity of length bias: smaller models exhibit stronger bias. As shown below, when concatenating five error-free segments, the Large variant produces roughly three times the predicted error score of the XXL variant:
>
> | Passage Index | 1 | 2 | 3 | 4 | 5 |
> | :--- | :--- | :--- | :--- | :--- | :--- |
> | Large | 0.00 | 1.41 | 2.49 | 3.38 | 4.16 |
> | XL | 0.00  | 1.02 | 1.61 | 2.24 | 2.61 |
> | XXL | 0.00 | 0.55 | 0.80 | 1.26 | 1.47 |
>
> ***Experiment #2***: Bias in Translation Text Length
>
> We also repeat the experiment from Section 3.4, which examines bias in translation text length. Recall that an unbiased model should assign roughly equal preference to two translations of the same quality but different lengths (i.e., ~50% preference for the shorter translation). In this setting, we again find that length bias is more pronounced in the smaller model, while the XL and XXL variants behave similarly, suggesting that for such use cases, increases in model size beyond a certain capacity yield diminishing returns for mitigating length bias:
>
> | Length Difference | 2.5 | 5 | 7.5 | 10 | 12.5 | 15 |
> | :--- | :--- | :--- | :--- | :--- | :--- | :--- |
> | Large | 57.2 | 57.3 | 57.3 | 57.7 | 59.6 | 62.5 |
> | XL | 53.5 | 53.6 | 53.4 | 54.4 | 55.9 | 56.5 |
> | XXL | 54.3 | 54.0 | 53.4 | 54.0 | 56.8 | 57.8 |
>
> Taken together, these results indicate that while increasing model capacity helps reduce length bias, substantial bias remains even in the largest publicly available MetricX-24 models, underscoring the need for model-agnostic mitigation strategies such as the one we propose. We will add these analyses to the Appendix section of the revised draft.
>
> **W3: Cause of Length Bias**
>
> We note that Section 4.1 of our paper analyzes the underlying cause of the observed length bias. Specifically, we conclude that the bias arises from skewed distributions of output scores in the models’ training data (i.e., training data biases), as evidenced by the training data statistics. Section 4.2 then shows that length normalization helps mitigate this issue by correcting the skew in the relative frequencies of output scores.
>
>
> **W4: Differential Analysis**
>
> Thank you for the insightful comment. The main reason we did not provide such a comparison is that the focus of our paper is to reveal the existence of length bias in state-of-the-art QE evaluation metrics. However, each metric is built on a different underlying model architecture, making a fair comparison difficult. For example, MetricX-24 is based on the T5 model, which cannot be used for LLM-as-a-Judge methods. We currently view a unified, architecture-controlled comparison as promising future work, and we plan to explore this direction in subsequent iterations of the project once such metrics emerge.

---

### Official Review · Reviewer_whPb · 2025-11-05

**Soundness:** 3
**Presentation:** 3
**Contribution:** 3
**Rating:** 8
**Confidence:** 2

**Summary:**

The paper investigates length bias in machine translation Quality Estimation (QE) models, which assess translation quality without references. The authors find that both traditional regression-based models like MetricX and COMET and large language model judges such as AutoMQM tend to give lower scores to longer translations, even when they are error-free.

Using data from ten language pairs in the WMT24++ dataset, they show that QE scores consistently decline with text length, and that models often prefer shorter outputs when comparing translations of equal quality. They claim that this effect stems from a lack of long, clean examples in the training data.

To reduce this bias, the authors propose two methods: training models to predict error density instead of raw errors and combining QE outputs with reference-based metrics when available. They find that these methods lessen the bias but do not remove it completely.

Overall, the paper provides a good empirical investigation of length bias in quality estimation. It shows that this bias can affect evaluation results and downstream processes such as reranking and RLHF. The proposed methods offer practical approaches to reduce the problem and improve the robustness of QE models.

**Strengths:**

- The paper addresses a relevant and underexplored problem: length bias in quality estimation metrics and it proposes a clear and effective solution to mitigate this issue.

- The experiments cover a wide range of QE architectures, including both regression-based and LLM-based models. The evaluation spans ten diverse language pairs, showing that the bias is systematic and cross-lingual.

- The proposed method is supported by strong empirical evidence, demonstrating its practical effectiveness.

**Weaknesses:**

- The paper lacks human evaluation results, which would strengthen the findings and provide a clearer link between metric predictions and human judgment. I think this is a minor point as the paper evaluates using multiple automated metrics

- The authors cite related work addressing similar issues, but they do not include direct comparisons or benchmarks against those existing approaches. Some relevant works from outside the MT community - (https://arxiv.org/pdf/2407.01085v5, https://arxiv.org/pdf/2406.17744)

**Questions:**

check weaknesses

---

> ### Author Response · Authors · 2025-11-25
>
> Thank you for your valuable and detailed feedback, as well as for your kind appreciation of our paper. We address your comments below.
>
> **W1: Human Evaluation**
>
> Thank you for noting this. We would like to clarify that WMT24++, the dataset used in our experiments, includes human evaluation results. As stated in Section 2.2 of the WMT24++ paper (https://arxiv.org/pdf/2502.12404), the reference translations and post-edits were produced by professional translators. Given that we select segments labeled as error-free for our experiments, the corresponding human judgments for these segments are naturally error-free as well.
>
> Furthermore, although the translations and post-edits were performed at the segment (paragraph) level, the translators were provided with the corresponding document context to aid in their understanding of the English text. Therefore, concatenating these segments into longer paragraphs is unlikely to introduce cross-segment fluency errors. Under these assumptions, we show that the examined QE metrics incorrectly predict increasing numbers of errors as the input length grows, even though human evaluation indicates no change in translation quality. We therefore conclude that this discrepancy reveals that QE metrics exhibit length bias. We will add this clarification to the dataset introduction section in the revised draft.
>
>
> **W2: Related Works**
>
> Thank you for suggesting additional related work addressing length bias. We agree that these approaches are effective in their original domains; however, they can be less suitable for the MT QE setting for several reasons.
>
> For AdapAlpaca, the method assumes access to sufficient reference answers of varying lengths, whereas QE typically operates without diverse reference translations, making it difficult to categorize responses by reference length. Moreover, the method relies on the assumption that longer responses convey more useful information, which does not generally hold in machine translation, where increased length often reflects redundancy or over-translation rather than higher quality.
>
> For length-instruction methods, target length variation in MT is often linguistically justified (e.g., due to word-order differences or morphological density), so imposing uniform length constraints may degrade fidelity. In addition, QE evaluates output quality rather than performing constrained generation, meaning that “learning to comply with a length bound” does not directly address evaluator-side bias. Finally, applying such length-instruction fine-tuning would require constructing translation pairs with equal true quality but controlled length, which is operationally difficult across languages and domains.
>
> We will add the mentioned related work to the related work section in the updated draft.

---

### Author Response · Authors · 2025-12-03
**Thank you for the discussion period.**

We express our sincere gratitude to the reviewers for their insightful and constructive feedback.

We appreciate that the reviewers acknowledge that:

1. The paper addresses an underexplored yet important and practically consequential issue (all four reviewers).

2. The experiments span a wide range of QE architectures (Reviewer whPb), are conducted in a sound and thoughtful manner (Reviewers 5fA6, wgfj), and yield reliable, generalizable findings (Reviewer wBLz).

3. The proposed mitigation is supported by strong empirical evidence that validates its effectiveness and provides direct value for improving QE metric robustness (Reviewers whPb, wBLz).

Throughout the discussion period, we have carefully addressed and responded to the concerns and questions raised by the reviewers, summarized as follows:

1. In response to Reviewers whPb and wgfj, we clarified that WMT24++ includes human-verified, error-free segments produced with full document context, so concatenating them does not introduce new errors. Under these conditions, the QE metrics’ rising error predictions reflect inherent length bias rather than genuine quality degradation.

2. In response to Reviewer wBLz, we provided additional analysis on model size selection. We found that although increasing model capacity helps reduce length bias, substantial bias persists even in the largest publicly available MetricX-24 models, underscoring the need for model-agnostic mitigation strategies such as the one we propose. We also highlighted how different error types and severities influence the magnitude of length bias, as requested.

3. In response to Reviewer wgfj, we provided additional training details (datasets and hyperparameters) for the proposed length-normalized model to ensure better reproducibility.

4. In response to Reviewer wgfj, as acknowledged by the reviewer, we clarified that our work examines the presence and causes of length bias in QE metrics rather than proposing a new evaluation metric. Accordingly, we treat length normalization as a diagnostic tool for understanding the bias, not a full replacement metric, and thus meta-evaluation, given the availability of human evaluation in the dataset, is not strictly applicable.

5. In response to Reviewers whPb and 5fA6, we acknowledged their suggestions on writing and related work and will incorporate them in the revised draft.

We sincerely appreciate the area chair’s effort in hosting the discussion period. We hope that our contributions and responses will be taken into full consideration.

---

### Meta-Review · Area_Chair_81Ge · 2025-12-11

**Summary:**

Several reviewers point at substance and focus. While, in general the topic is well accepted, the reviewers state there is not enough analysis, suggest more experiments, suggest the framing is confusing (what is the paper about? Why this paper is exciting?) and calls for more works.

**Reviewer Concerns:**

The paper did additional experiments, explained comments about human annotation etc. Still, it is not clear that this are enough changes and that the overall paper is now in a clear standalone and ready form. Also, related works were addressed in the fact that missing works found by the reviewers were added, but I advise the authors to do a better job in related work rather than just add the works someone happened to pick as an example.

**Reviewer Scores:**

That is not a fair, relevant or meaningful question. I protest the way this was all handled.
A Reviewers are not here, and ToM is weak, at least mine and the one literature study. I will not try to predict people.
B Scores are, anyway, a weak signal of interest; a paper should not be accepted or rejected just based on it. An AC's job is to look at the specific weaknesses and translate them into a recommendation.
C There are about 100 pages of discussions for me to read overall, in addition to the discussions I monitored and were just replaced, this is beyond my personal ability to do fairly. I did my best effort.

---

### Decision · Program_Chairs · 2026-01-26

Reject